# Leveraging hyperspectral remote sensing and foundation models for greenhouse monitoring

Alberto Costa Nogueira Jr.
albercn@br.ibm.com
IBM Research

Devyani Lambhate
devyani.lambhate1@ibm.com
IBM Research

João L. S. Almeida
joao.lucas.sousa.almeida@ibm.com
IBM Research

Levente Klein
kleinl@us.ibm.com
IBM Research

Maciel Zortea
mazortea@br.ibm.com
IBM Research

Ranjini Bangalore
rangurup@in.ibm.com
IBM Research

Ronald Albert
ronaldalbert@ibm.com
IBM Research

Thomas Brunschwiler
tbr@zurich.ibm.com
IBM Research

## Abstract

Greenhouse gas (GHG) emissions have become a critical environmental concern, significantly impacting global climate change. Given the global coverage and multiple gas species that may be emitted simultaneously, a comprehensive strategy is required to capture the spatial-temporal variation and subtle atmospheric photochemical reactions. Current approaches to GHG detection rely on detecting a single gas species using broad multispectral bands in short-wave infrared but disentangling different chemical gases and their ratios is not achievable. Hyperspectral imaging, a remote sensing technology, has emerged as a powerful tool for detecting and monitoring GHG concentrations in the atmosphere. While traditional retrieval GHG algorithms, such as Matched Filters are helpful, they are prone to false detection and typically require postprocessing, which may require manual interpretation. To address these challenges, this position paper aims to advocate for exploring foundation models, a self-supervised artificial intelligence (AI) model to detect GHG from multimodal data (optical, LiDAR, and weather observations/forecasts) that reconstruct missing data but at the same time captures the physical and chemical processes in the atmosphere across different spatial and temporal scale. In this context, we present a conceptual outline of potential GHG foundation models built on multimodel data that leverage spatial-temporal-spectral reconstructions and learn from each data modality. We will discuss the challenges associated with GHG FM implementation.

## Keywords

Greenhouse gas, machine learning, remote sensing.

**ACM Reference Format:**
Alberto Costa Nogueira Jr., Devyani Lambhate, João L. S. Almeida, Levente Klein, Maciel Zortea, Ranjini Bangalore, Ronald Albert, and Thomas Brunschwiler. 2024. Leveraging hyperspectral remote sensing and foundation models for greenhouse monitoring. In *Proceedings of Fragile Earth workshop.* ACM, New York, NY, USA, 4 pages.

*Fragile Earth workshop, August 26, 2024, Barcelona, Spain*
© 2024

## 1 Introduction

The primary GHGs in the atmosphere are carbon dioxide (CO2), methane (CH4), and nitrous oxide (N2O), which together account for about 95% of all GHG emissions. These gases absorb infrared radiation and then re-emit them, causing the Earth's surface to warm up, a phenomenon known as the greenhouse effect. The warming potential of greenhouse gases (GHGs) on Earth's climate system is well understood but current measurement and sensing technologies can not provide a complete understanding of the spatio-temporal evolution of emission sources, absorption sinks, quantification of the emission and absorption, and photochemical reactions for all gases [12].

GHG emission sources can be categorized as point, area, or mobile sources and their emissions are either intermittent or persistent. In general, point sources with persistent emissions are the easiest to detect as they can be measured multiple times across different detection modalities. Identifying and quantifying these major sources is crucial for building accurate GHG inventories and designing, prioritizing and validating GHG reduction and mitigation actions.

Ground based measurements (e.g. local sensors like CO2 weather station measurements) and remotely sensed measurements (satellites, airborne vehicles) of GHG emission/absorption are commonly used to identify the sources/sinks and quantify the emission and absorption of GHG. While ground based measurements are accurate and give the surface level GHG fluxes, they are not scalable. The recent launches of several satellites (including OCO2, GHGSat, Sentinel 2 and 3, and Sentinel 5p) enable GHG emissions to be monitored and quantified at an unprecedentedly high spatial (20 meters to 7 kilometers) and temporal (1-16 days) resolution. However, satellite measurements are occluded by clouds, aerosols, and are limited sparse spatial and/or temporal (high revisit times) and/or spectral resolutions.

Specifically, for GHG monitoring, fine spectral resolution is key as greenhouse gases near the Earth's surface attenuates the reflectance of the sunlight bounced from the Earth's surface in specific wavelengths. For instance, in the case of CH4, this happens primarily in the short-wave infrared region within the wavelength ranges of approximately 2.2–2.4 $\mu$m, and for CO2, it is around 1.4–1.6 and 1.9–2.1 $\mu$m [13]. The fine spectral resolution resolved by hyperspectral imaging instruments allows the detection of subtle attenuations

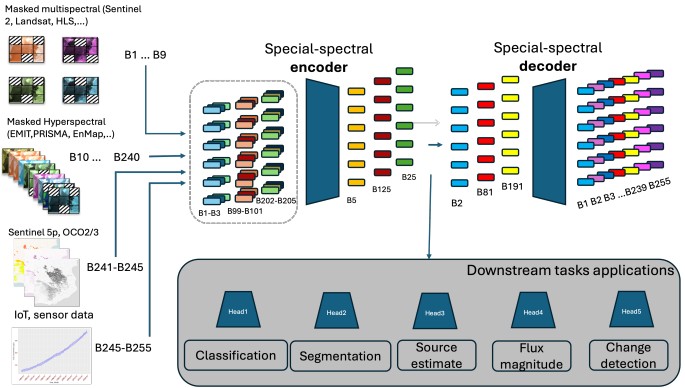

**Figure 1: AI framework for detection of GHG sources and estimation of GHG emissions.**

in these spectral responses, which translate into absorption features in the spectral response measured for each pixel.

AI based approaches such as physics based learning models, and deep learning models have been investigated to learn representations and patterns from remote sensed data, including hyperspectral data.

Foundation models are designed as broad-spectrum models, capable of managing various tasks, and are trained on extensive data quantities. These models can discern intricate patterns and relationships in data, making them suitable for image analysis, language processing, and decision-making. Their distinct advantage lies in their more flexible architecture and their training on diverse tasks and datasets, especially taking advantage of unlabeled data. This results in foundation models learning general data representations, applicable across numerous tasks and domains. Self-supervised learning, used in foundation models, that can learn latent space representations of missing data is ideally positioned to capture both spatial-temporal-spectral data that may be blocked or is missing in the datasets. While spectral reconstructions may be easier in optical satellites like Sentinel-2 or Landsat, the close proximity of bands and sharp spectral absorption of GHG gases, positions the spatial-spectral FM models to identify GHG sources and quantify magnitudes. The GHG FM model proposed has distinct advantages compared to current single modality GFM models [5]:

- Accept multi modal data input in the form of raster images as well as time series and point measurements data
- The model will learn spectral information from hyperspectral bands as well as translation of hyperspectral to multispectral data
- Enforces the physics and chemistry constraints related to energy fluxes, symmetry, and first principle physics laws.

In this paper, we discuss a conceptual outline of a potential solution using multi modal/spectral/temporal foundation models to monitor GHG emissions. For this, we propose to explore foundation model technology [7, 11] and fuse it with real-time satellite observations, and AI-driven approaches (i.e., downscaling using a weather foundation model and inverse modeling using a Physics-based approach available on SimulAI [6] to super-resolve emission sources

and sinks down to the scale of the GHG emission infrastructure. A high-level sketch showing examples of input data and its use in foundation models is shown in Figure 1.

## 2 Related work

### 2.1 Methane and carbon dioxide retrievals using hyperspectral imagery

The pixel's hyperspectral signature carries absorption patterns that enable a retrieval algorithm to identify the presence of gases. An example of such an algorithm is Matched Filters [4], a signal processing technique used to identify specific patterns in a signal. They operate by comparing the input signal with a predefined reference signal which contains the pattern of interest. In the context of GHG identification, linearized Matched Filters compare the shape of the pixel's response differing from its background (which needs to be estimated) with the shape of unit absorption spectra of the gas of interest, obtained from radiative transfer simulations [16]. The output of the filter, referred to as mixing ratio length in units of ppmm (parts per million meter), represents the thickness and concentration within a volume of equivalent absorption [16]. Due to its simplicity, Matched Filters have been successfully applied to detecting CH4 and CO2 using hyperspectral imaging, either from airborne or spaceborne instruments [3, 16]. However, these detections require further processing, often involving human intervention, to distinguish legitimate sources from false detections caused by land features that provide a similar match.

The scientific community is actively investigating alternative methods to enhance automation in this process. Recent examples include the application of deep learning models that utilize the response of Matched Filters as input [15], and transformer architectures tailored to wavelengths sensitive to methane [9].

### 2.2 Foundation models

With the proven success of foundation models in natural language processing and computer vision, there is an interest in applying the same approaches to self-supervised models for geospatial applications [7]. Self-supervised learning (SSL) allows foundation models to benefit from the massive volume of unlabeled geospatial data, learning useful representations and correlations. These pre-trained models can then be quickly adapted to specific tasks (usually called downstream tasks) using a small volume of labeled datasets, reducing the need for manual annotations. Given the very dynamic patterns and processes in Earth Observations, many of the current Foundation Model architectures are single modality models without considering the complementary information carried by multimodel datasets and the possibility of reconstructing spatial, temporal, and spectral gaps.

## 3 Hyperspectral data: EMIT products

EMIT is an imaging spectrometer designed to measure reflected solar radiation across 285 distinct wavelengths, spanning the visible to shortwave infrared range (381 to 2493 nanometers). Launched in mid-2022, the EMIT instrument is installed on the International Space Station and acquires hyperspectral images from Earth's surface regions within ±52° latitude. The open data portal includes the

"high-confidence research-grade methane plume complexes from point source emitters."[1] This product contains manually delineated individual or overlapping methane plumes, along with pixel concentration in ppm m derived from radiance images using Matched Filters parameterized for each EMIT scene (atmospheric water vapor modeled by the primary EMIT mission, path length, viewing geometry), along with uncertainty data. The shape of the methane plumes is determined by three independent experts who visually inspect the Matched Filter results, review detection, and manually outline the plumes [1].

## 4   Challenges and Considerations

In principle, GHG FM could be trained to detect multiple gases simultaneously, meaning that the image patches could pass through an encoder-decoder architecture only once. Conversely, Matched Filters would detect a single gas at a time. Additionally, GHG FM could optionally incorporate other data sources as input to the detection process, such as weather data to verify if plumes are dispersing along prevalent wind conditions to reduce false negative detection. In our proposed framework the following distinguishing elements are relevant for the FM model:

- Exploring the multimodal nature of the geospatial data consisting of different modalities: optical, LiDAR, and weather
- Learning between various modalities of detection or between different wavelengths of the hyperspectral images
- Integration of physics into models through loss function constraints to create enhanced representations.

Below we point to some of the foreseen challenges.

### 4.1   Data availability and quality

We believe that open access to EMIT imagery (EMIT-L1B-L2A, EMIT-L2A-RFL) and its derived products (L2B-CH4PLMMETA, EMIT-L2B-CH4ENH) are valuable to fit the foundation models. We can sample patches from thousands of images globally and use a pretext task such as reconstructing masked parts of the image, to perform semi-supervised learning on its massive dataset. A pretext task in foundation models is a simple, yet challenging task, designed to learn useful representations of the input data. The goal of pretext tasks is to create a learning environment where the model can learn to perform well without explicit supervision on the target task [17]. Pretext tasks can take various forms, such as image classification, object detection, or semantic segmentation.

However, if data is sampled over large areas, and randomly, the impact of using EMIT imagery that mostly is unlikely to include point sources of greenhouse gases (GHG) during the self-supervised learning might be a concern if the subsequent task is GHG monitoring.

While the gas enhancement output of Matched Filters, which relies on physical principles, relates to gas concentration, how to get concentration for training FM remains an open issue. One alternative would be to rely on creating synthetic datasets, following approaches similar to [8] or [14, 18].

[1]https://earth.jpl.nasa.gov/emit/data/data-portal/Greenhouse-Gases

### 4.2   Model interpretability

Deep learning/FM models can be quite challenging to interpret, making it difficult to grasp the underlying rationale behind their outputs. Imposing physics constraints in loss functions through scientific machine learning principles [6] such as mass, energy, and momentum conservation [2] produces more reliable and interpretable models. However, adding physical knowledge to neural network training may decrease convergence rates since the loss minimization becomes stiffer depending on the complexity of the embedded physics. The higher the fidelity of the physical description in the loss function, the slower the convergence of the optimization problem. Interpreting Matched Filters in this context might be relatively simpler due to the fewer parameters of the model.

### 4.3   The Encoding of Spectral Information

As discussed in Section 2.1, traditional methods for detecting GHG on hyperspectral images typically involve constructing filters, that exploit the relationship between bands to enhance the GHG signal. Therefore, a successful model for effectively monitoring greenhouse emissions is highly dependent on spectral information. In [10], the authors propose a strategy for self-supervised training focused on spectral modeling, it works by masking out random bands in the original data and trying to predict it in an MAE fashion. The best way to implement pre-training of a foundation model capable of efficiently encoding spectral information is still unclear and an open and interesting object of research.

### 4.4   Example of applications

*4.4.1   CO2 estimates.* We have conducted preliminary experiments to assess a model's effectiveness in estimating $CO_2$ from hyperspectral data. In these experiments, we used a simple Random Forest regressor and a Vision transformer model pre-trained on The Environmental Mapping and Analysis Program (EnMAP) hyperspectral sensor. These experiments were conducted on a small dataset of 144 (224×224 dimensional images). We used OCO2 and OCO3 data as labels and corresponding EnMAP data as input. Because of the difference in revisit time of EnMAP (27 days) and OCO2/3 satellites (16 days), the two datasets are matched if they are within 15 days data acquisition interval for the OCO2/3 and EnMAP data. The model provides an RMSE of 2.8 ppm on a range of 10 ppm using a spatial-spectral Vision transformer and an RMSE of 0.49 ppm using a Random Forest model. The results clearly show that hyper-spectral data can be used to estimate $CO_2$. The higher RMSE from the Spatial-spectral Vision transformer might be because of fewer labeled samples (the 144 images used are sparse and contain some repetition in the OCO2/3 data because of the coarser spatial resolution of OCO2/3 compared to EnMAP).

*4.4.2   CH4 Plume segmentation.* Figure 2 illustrates a preliminary example of binary CH4 plume segmentation using EMIT-L1B-RAD imagery. For this, we randomly sampled ≈23k image patches of size 64×64 pixels, originating from 540 EMIT images for training, and disjoint sets of 68 images for validation, and 68 for testing, randomly split from acquisitions from August 2022 to February 2024. All images were known to have methane plumes as delineated in the companion EMIT-L2B-CH4PLMMETA metadata. To sample

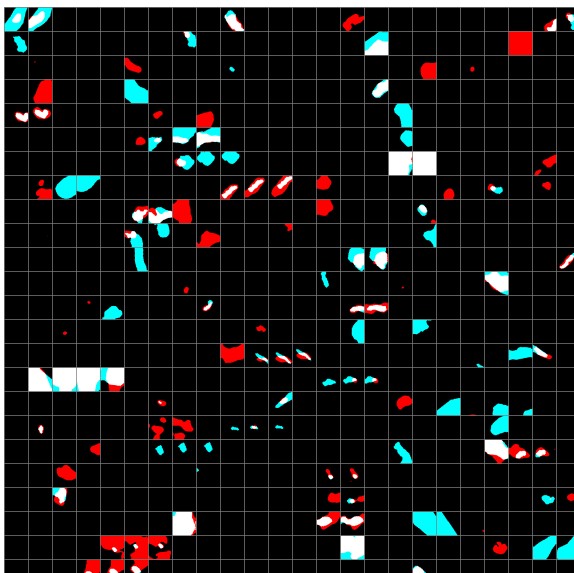

**Figure 2: Detail of the agreement between EMIT reference plume masks and the corresponding segmentation obtained using U-Net applied to {mag1c,RGB} inputs. Each of the 576 images in the 24×24 grid corresponds to a patch with dimensions of 64×64 pixels randomly sampled from the test set, roughly 3.8 km×3.8 km on the ground. Correctly predicted methane pixels (true positives) are shown in white, correctly predicted non-methane (true negatives) in black, underprediction (false negatives) in cyan, and overprediction (false positives) in red.**

positive and negative cases for the presence of plume, we randomly sampled patches centered in 10 coordinates inside the reference plume masks and 50 elsewhere in each image. From each EMIT image, we retained the output of the "mag1c" [4] implementation of the Matched Filter[2] and include the RGB channels to form the 4-channel input (like in [15]) to the segmentation model, which is a U-Net with a "resnet50" decoder. We trained a segmentation model for 40 epochs, using Adam optimizer with an initial learning rate of 0.001 and final 0.0005, minimizing the average Dice loss over batches of 128 patches. Preliminary results suggest that it is challenging to match the plumes delineated by the polygons available in the EMIT-L2B-CH4PLMMETA product used as a reference. In this task, we achieved a pixel-wise F1 score of 0.52 in the test set. Optimizing how the hyperspectral bands are used may lead to improved results, instead of the ad hoc selection of mag1c output and RGB channels.

## 5 Conclusion

GFM offers significant potential for improving greenhouse monitoring and management practices. By addressing challenges related to data quality, and model interpretability we can harness the power of these technologies to identify emission hot spots and to inform mitigation strategies. Foundation models have been little explored in the context of retrieving GHG plumes and quantifying emissions,

and we believe that combining multimodal sensing with spectral learning and enforcing AI models using physics constraints can advance GHG detection.

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
