# OpenReview forum: "Leveraging hyperspectral remote sensing and foundation models for greenhouse monitoring"
_KDD.org/2024/Workshop/Fragile_Earth — Fragile Earth FullPresentation_

### Official Review · Reviewer_TypW · 2024-07-13
**Review of Leveraging hyperspectral remote sensing and foundation models for greenhouse monitoring**

**Rating:** 6
**Confidence:** 4

**Review:**

Summary:

In this paper, the authors propose a conceptual outline of a potential solution using multi modal/spectral/temporal foundation models to
monitor GHG emissions and show that GFM offers significant potential for improving greenhouse monitoring and management practices.

Strengths:
+ The paper is well-written and generally well structured.
+ Related work is covered thoroughly and differentiates the contributions of the paper well.
+ Examples of real-world applications are promising and interesting.

Weaknesses:
- The authors can apply state-of-the-art machine learning algorithms in the experiments. For instance, to estimate CO2, instead of using random forest, other neural network-based models can be used and compared as well.

---

### Official Review · Reviewer_4f9f · 2024-07-16
**Review for "Leveraging hyperspectral remote sensing and foundation models for greenhouse monitoring"**

**Rating:** 7
**Confidence:** 4

**Review:**

Summary

This paper discusses the challenges and prospects of adopting foundational models to monitor greenhouse gas emissions using multi-modal data.

Strengths

- The motivation for this work is clear, with the authors communicating the gap their proposed foundational models could fill compared with the currently available techniques and methodology.
- The inclusion of real-world applications and their preliminary results were a great touch.
- A significant strength of the proposed GHG FM model is that it accepts multi-modal data. This is a significant development not only for GHG emissions but for other domains with limited data that require time series or point monitoring. Biodiversity conservation is an example, with data gathered from many sources (camera traps, remote sensing, field surveys, etc) frequently not communicating with one another.

Weaknesses/Feedback

- The authors also fairly present the challenges in interpreting complex computational models in this domain. In the examples provided in 4.4, it would be useful to discuss the limitations of the results provided. For example, CO2 estimates in 4.4.1 were predicted on a relatively small dataset. There is a lot of noise in computational methods when insufficiently sized datasets are used, leading to inconsistent results. -- A more thorough analysis using different dataset sizes or samples to understand the variance in outputs would be prudent.

---

### Decision · Program_Chairs · 2024-07-24

Accept (Full Presentation)